# Activation of the Mitochondrial Unfolded Protein Response: A New Therapeutic Target?

**DOI:** 10.3390/biomedicines10071611

**Published:** 2022-07-06

**Authors:** Juan M. Suárez-Rivero, Carmen J. Pastor-Maldonado, Suleva Povea-Cabello, Mónica Álvarez-Córdoba, Irene Villalón-García, Marta Talaverón-Rey, Alejandra Suárez-Carrillo, Manuel Munuera-Cabeza, Diana Reche-López, Paula Cilleros-Holgado, Rocío Piñero-Pérez, José A. Sánchez-Alcázar

**Affiliations:** 1Centro Andaluz de Biología del Desarrollo (CABD-CSIC-Universidad Pablo de Olavide), Centro de Investigación Biomédica en Red: Enfermedades Raras, Instituto de Salud Carlos III, 41013 Sevilla, Spain; jmsuariv@upo.es (J.M.S.-R.); carmen.pastor-maldonado@student.uni-tuebingen.de (C.J.P.-M.); spovcab@upo.es (S.P.-C.); malvcor@upo.es (M.Á.-C.); ivilgar@acu.upo.es (I.V.-G.); mtalrey1@alu.upo.es (M.T.-R.); asuacar1@alu.upo.es (A.S.-C.); mmuncab@upo.es (M.M.-C.); dreclop@alu.upo.es (D.R.-L.); pcilhol@alu.upo.es (P.C.-H.); rpieper@alu.upo.es (R.P.-P.); 2Centro Andaluz de Biología del Desarrollo (CABD), Consejo Superior de Investigaciones Científicas, Universidad Pablo de Olavide, Carretera de Utrera Km 1, 41013 Sevilla, Spain

**Keywords:** mitochondria, homeostasis, unfolded protein response, proteostasis, therapeutic target, mitochondrial diseases, neurodegeneration, lifespan, aging, heart diseases

## Abstract

Mitochondrial dysfunction is a key hub that is common to many diseases. Mitochondria’s role in energy production, calcium homeostasis, and ROS balance makes them essential for cell survival and fitness. However, there are no effective treatments for most mitochondrial and related diseases to this day. Therefore, new therapeutic approaches, such as activation of the mitochondrial unfolded protein response (UPR^mt^), are being examined. UPR^mt^ englobes several compensation processes related to proteostasis and antioxidant mechanisms. UPR^mt^ activation, through an hormetic response, promotes cell homeostasis and improves lifespan and disease conditions in biological models of neurodegenerative diseases, cardiopathies, and mitochondrial diseases. Although UPR^mt^ activation is a promising therapeutic option for many conditions, its overactivation could lead to non-desired side effects, such as increased heteroplasmy of mitochondrial DNA mutations or cancer progression in oncologic patients. In this review, we present the most recent UPR^mt^ activation therapeutic strategies, UPR^mt^’s role in diseases, and its possible negative consequences in particular pathological conditions.

## 1. Mitochondria and Homeostasis

Homeostasis is defined as any self-regulating process by which biological systems tend to maintain stability while adjusting to conditions that are optimal for survival. In short, homeostasis is the capacity of organisms to adapt to external or internal changes. The cellular responses must be tightly regulated and fast in order to allow for adaptation to environmental or internal changes and the maintenance of cellular survival. These mechanisms are the basis of cellular homeostasis, organized at several molecular levels, and require a perfect coordination: epigenetic remodeling [1], transcriptional responses [2], post transcriptional modifications [3], organelle reorganization [4,5], osmotic control [6], and protein regulation [7], among many other things, determine if the cells will survive or die depending on whether their responses are flexible enough to adapt.

Mitochondria are a key organelle for cell homeostasis [8]. They are vital compartments for every nucleated cell, given that energy production through oxidative phosphorylation (OXPHOS) is their main function. Additionally, mitochondria are necessary for steroid and heme biosynthesis [9], numerous metabolites’ production [10], ion regulation [11,12], redox signaling [13], programmed cell death [14], and innate immunity [15]. Because of mitochondria’s importance for cell homeostasis, mitochondrial dysfunctions cause a great variety of diseases which can affect almost all the tissues in the body. Mitochondrial dysfunction is generally associated with reactive oxygen species (ROS) overproduction, being closely correlated and present in all mitochondrial diseases and aging [16]. Due to the high ROS production of these organelles, their components tend to degenerate, resulting in underperforming mitochondrial function [17]. On the other hand, mutations in the mitochondrial or nuclear genome produce aberrant proteins that contribute to ROS overproduction and functional loss [18]. These alterations might not only lead to mitochondrial diseases but also to the development of neurodegenerative diseases [19], aging [20], fertility loss [21], cancer [22], diabetes [23], and a vast spectrum of related pathologies.

In order to maintain a healthy mitochondrial network, cells must possess mechanisms to recycle and repair defective mitochondria. Mitochondrial quality control is an important aspect of cellular homeostasis, especially in postmitotic tissues, since it protects against the release of pro-apoptotic proteins, ROS production, and ineffective generation of ATP by damaged or aged mitochondria. The regulation of the mitochondrial life cycle and the maintenance of a robust functional network within cells are achieved through the maintenance of a precise balance between mitochondrial turnover and biogenesis [24]. Mitophagy, the selective degradation of damaged mitochondria by autophagy [25], is necessary for mitochondrial quality control and is the only known pathway to enable the turnover of whole mitochondrial genomes [26]. Mitochondria could be targeted for autophagic degradation for a variety of reasons, including basal turnover for recycling, starvation induced degradation, and degradation due to damage [25]. The core autophagic machinery is evolutionarily highly conserved, and the signaling cascades regulating autophagy are common to both bulk and selective autophagic processes [27]. Damaged or depolarized mitochondrial fragments are engulfed by autophagosomes, double-membraned organelles that are central to macroautophagy. Autophagy initiation depends on the recruitment and maturation of microtubule-associated protein-1 light chain-3 (LC3) to the autophagosomal membrane. In selective autophagy processes, such as mitophagy, the recognition of the cargo is mediated by cargo receptors that interact with matured LC3 (LC3-ll) through the LC3-interacting region (LIR) domain. Acidification and degradation of the cargo follow the formation of autolysosomes by fusion of the autophagosomes with lysosomes. Under physiological conditions, mitochondrial removal and replenishment reaches an equilibrium, allowing the maintenance of a constant mitochondrial volume. This balance is established according to tissue type, energy demand, activity level, stress, mutations, and abundance of nutrients [28].

Contrary to mitophagy, the mitochondrial unfolded protein response (UPR^mt^) is a mechanism aimed to preserve or repair damaged mitochondria. UPR^mt^ is responsible for maintaining mitochondrial proteostasis via mitochondrial activation of a transcriptional program in the nuclear DNA [29]. This compensation system can be divided into three main pathways: activation of (1) chaperones, which boost refolding of misfolded proteins to restore them to their functional conformation and assist the folding of newly synthesized proteins; (2) proteases that are able to degrade aberrant proteins or aggregates; and (3) an antioxidant system that palliates ROS overproduction [30]. Although human UPR^mt^ is not fully understood, it is gaining relevance in a variety of physiological processes on top of its canonical function, such as ageing, oxidative stress resistance, hematopoietic stem cell maintenance, glycolysis, antibacterial immunity, coenzyme Q biosynthesis, and mitochondrial fission [31,32]. Loss of mitochondrial proteostasis is the main UPR^mt^ inducer. Accumulation of damaged proteins exceeding the protein-processing capacity of the chaperones and proteases in mitochondria would activate UPR^mt^, for instance [33]. Additionally, factors interfering with mitochondrial function promote UPR^mt^ induction. Examples of these are the inhibition of complex I by rotenone [34], bacterial toxins [35], knockdown of quality control proteins [36], or generation of excess ROS by paraquat [37].

Nonetheless, if the UPR^mt^ is unable to repair the mitochondrial damage, elimination of the entire mitochondria is promoted. Mitophagy is likely the last-resort response because it requires the cell to replace the organelles. The challenge remains to delineate the exact signaling components and the temporal characteristics of these two processes in the recovery of the mitochondrial network [38]. Finally, if mitophagy is impaired or the damage persists, cells undergo senescence and/or apoptosis [39] (Figure 1).

In this review, we focus on the growing interest of UPR^mt^ induction as potential therapy for several mitochondria-related diseases and its possible consequences.

## 2. What Is the UPR^mt^?

Although UPR^mt^ was discovered in mammalian cells [40,41], it has been thoroughly studied in *C. elegans* [42]. The stress-activated transcription factor 1 (ATFS-1) was identified in *C. elegans* as a key regulator of the UPR^mt^. ATFS-1 is normally imported into mitochondria, but during mitochondrial dysfunction, a percentage of ATFS-1 accumulates in the cytosol and then traffics to the nucleus, where it induces transcription of mitochondrial chaperones, proteases, and antioxidants [42,43]. Consistent with mediating a protective transcriptional response, worms lacking ATFS-1 incur respiratory defects during mitochondrial stress and higher susceptibility to mitochondrial perturbations [35,43]. This process is evolutionarily conserved [44], and there are three key protein regulators of human UPR^mt^: Activation Transcription Factor 5 (ATF5), Activation Transcription Factor 4 (ATF4), and C/EBP homologous protein (CHOP) [45,46]. There is still no consensus about how these proteins interact with each other or of their exact role in UPR^mt^ regulation. What is indeed known is that all three are upregulated after mitochondrial damage, either by exposure to rotenone, mtDNA depletion, or mitochondrial proteins over-aggregation [45,47,48]. All three transcription factors are also involved in the integrated stress response (ISR), which is a conserved adaptive response that is activated by a wide variety of stressors [33]. In fact, in mammals, the ISR acts as an essential precursor of UPR^mt^ activation. Sensing, surviving, and adapting to mitochondrial dysfunction depends on attenuated protein synthesis, privileged translation of transcription factors, and the activity of the ISR eucaryotic initiation factor 2 alpha (eIF2α) kinases [47].

Even though each of these transcription factors is implicated in promoting transcription of genes required to overcome mitochondrial dysfunction, it is unclear whether they function individually or in concert. Studies show that they regulate the expression of one another [48]. There are many examples of their role in restoring mitochondrial function: HeLa cells lacking a functional copy of ATF4 failed to upregulate several mitochondrial enzymes and exhibited a reduction in ATP-dependent respiration [49]; gene expression analysis of human and mouse tissues revealed a tight correlation between ATF4 induction and UPR^mt^ responsive genes [45]; worms lacking ATFS-1 failed to induce chaperones’ expression in response to mitochondrial dysfunction were rescued by ATF5 expression but not ATF4 [44]; knockdown of ATF5 in HEK293 cells dampened induction of UPR^mt^ responsive genes [50]; and elevated transcript levels of ATF4 and ATF5 were detected in mice harboring a mutation in the mitochondrial DNA helicase [51]. In addition, global transcriptomic analyses have validated the presence of CHOP-binding elements in many UPR^mt^ gene promoters, while also revealing abundant ATF4-binding motifs [49].

In general, the most accepted pathway describes that ATF5 shuttles between the mitochondria and nuclei and induces the UPR^mt^ to promote cell proliferation and mitochondrial functional recovery under mitochondrial stress. In addition, mitochondrial dysfunction upregulates CHOP, which can act as transcription factor that binds to the promoters of UPR^mt^-related genes, thereby inducing the expression of mitochondrial chaperones, proteases, and antioxidants [52]. Furthermore, ISR induction leads to the increased translation of ATF4, which activates many genes, including CHOP [53]. Nonetheless, recent studies expose that there are many other proteins implicated in the UPR^mt^, such as the whole sirtuin family [54]. For an in-depth description of the UPR^mt^ molecular pathways, we recommend the recent reviews from Yun et al. [55], Naresh et al. [47], and Anderson et al. [33].

From a clinical point of view, UPR^mt^ induction could be linked to a novel concept derived from an old statement: mitohormesis. Hormesis is defined as any adaptive mechanism exhibiting a biphasic dose response. Mitohormesis is a broad and diverse cytosolic and nuclear response that can be triggered by any of a variety of insults leading to mitochondrial stress. Interestingly, this response seems to induce a wide-ranging cytoprotective state, resulting in long-lasting metabolic and biochemical changes. Remarkably, rather than being harmful, these changes may reduce susceptibility to disease, as well as potentially determine lifespan [55]. In the context of mitohormesis, the activation of the UPR^mt^ might provide biologically important short- and long-term adaptation against several pathologies. In fact, there is a growing body of evidence suggesting that activation of the UPR^mt^ might be an important determinant of lifespan [56]. The clearest example of hormesis is physical exercise. From an objective point of view, physical exercise is considered a form of stress (ROS production, lactic fermentation, and muscle fibers’ damage), yet physical exercise by itself increases the production of antioxidant proteins. In fact, this response was not observed in individuals taking a combination of vitamin C and vitamin E. The salutary benefits of exercise appeared to be inhibited in subjects given antioxidant supplements [57]. Therefore, the mild oxidative stress caused by exercise promotes cells to produce antioxidants which will protect the organism; however, antioxidants will block the oxidative signal making the organism prone to oxidative damage in the long term. Ranging down to mitochondria, mild and isolated stress could lead to the activation of several compensation mechanisms, optimize mitochondrial function, and therefore improve cellular fitness. A schematical representation is shown in Figure 2.

However, the contribution of particular genes in the UPR^mt^ response pathway should be carefully evaluated, taking into account that many of them are constitutive genes that are required for the maintenance of basic cellular function, and their up- or downregulation may produce numerous changes in the cells, including the mentioned UPR^mt^.

## 3. Mitochondrial Diseases

Mitochondrial diseases encompass a broad spectrum of muscular and neurodegenerative disorders, both chronic and progressive, caused by mutations in nuclear (nDNA) or mitochondrial (mtDNA) DNA [58]. The prevalence of these diseases has been established at 1:5000 [59]; however, this number is increasing due to the standardization of whole-exome sequencing [18]. Most OXPHOS disorders in children are a consequence of the mutation of nuclear DNA, and they are transmitted as autosomal recessive traits, usually with severe phenotypes and a fatal outcome. Among the maternally inherited pathogenic mtDNA mutations, more than 50% have been identified in genes encoded by mitochondrial transfer RNAs (mt-tRNA) (MTT genes). At first sight, it might seem paradoxical to induce a mild mitochondria stress in a general mitochondrial dysfunction context; however, recent studies present this strategy as a new alternative treatment for mitochondrial diseases.

Perry et al. showed that the tetracycline antibiotics family increased cell survival and fitness in MELAS cybrids and Rieske cells (Knockout Complex III mouse fibroblasts) under glucose restriction [60]. Specifically, doxycycline improved survival in wild-type cells treated with piericidin (complex I inhibitor) or antimycin (complex III inhibitor) during glucose deprivation. In addition to tetracyclines, the anti-parasitic agent pentamidine and the antibiotic retapamulin also scored positive on the screening in MELAS and mutant ND1 cybrid cells. They propose a “mitohormetic effect” by the induction of ATF4 (UPR^mt^) and p-eIF2α (ISR) as mechanism of action; however, in some cases, cell survival was independent of ATF4, suggesting the high complexity of this pathway. Suarez-Rivero et al. also demonstrated that tetracycline treatment boosts the production of UPR^mt^-related proteins and promotes the activation of pathways involving cAMP and cGMP, which might be implicated in mitochondrial compensatory mechanisms comprising sirtuins and chaperones’ activity [61,62,63]. Tetracycline treatment and the subsequent activation of UPR^mt^, the increased number of chaperones, and mitochondrial auxiliary proteins may promote the stability of a fraction of mutated mitochondrial proteins which would carry out their function to some extent [64]. All of these antibiotics have one factor in common: they are inhibitors of the mitochondrial translation at different degrees [65,66,67]. 

Given the bacterial origin of mammalian cells’ mitochondria and the fact that they conserve prokaryotic features such as the 55S or 60S ribosomes, it is understandable that mitochondria are exceptionally sensitive to antibiotics [68]. In this way, mitohormesis can be triggered by a partial inhibition of mitochondrial translation promoting a beneficial retrograde signaling response including the modulation of mitochondrial dynamics, the expression of nuclear and mitochondrial-encoded genes, the antioxidant response, stimulating mitochondrial function, and boosting cellular defense mechanisms that increase stress resistance [69].

Although the research of Perry et al. and Suarez-Rivero et al. is promising, antibiotic use in mitochondrial diseases is still in debate [70]. Several antibiotics can worsen the conditions of individuals bearing mtDNA mutations. Aminoglycosides, for instance, induce ototoxic hearing loss in subjects with mutations in the 12S rRNA gene [71]. Although aminoglycosides typically display specificity toward prokaryotes over eukaryotes, human mitochondrial ribosomes that have A1408 and G1491 at analogous positions exhibit higher resemblance to their bacterial counterparts. This similarity is likely responsible for some of the adverse effects shown by aminoglycosides [72]. 

It has recently been demonstrated that pterostilbene in combination with mitochondrial cofactors treatment activates SIRT3 and UPR^mt^ as compensatory mechanisms, as well as enhances sirtuins’ levels and mitochondrial activity in several cell models of mitochondrial diseases [73]. In contrast to long-term antibiotic treatment, pterostilbene is considered safe for human consumption and presents numerous well-known features, such as a prominent antioxidant activity and a high anti-inflammatory potential [74]. Furthermore, it has been reported to prolong lifespan in several animal models [75] due to its neuroprotective [76] and cardioprotective [77] properties. At the molecular level, pterostilbene activates sirtuins [75] and has been linked to AMP-activated protein kinase (AMPK) [78] and Nuclear factor erythroid 2-related factor 2 (Nrf2) by recent studies [79], suggesting that it could be a promising compound to maintain mitochondrial homeostasis.

## 4. Neurodegeneration

Neurodegenerative diseases are a widely heterogeneous group of disorders characterized by the progressive degeneration of the structure and function of the central and peripheral nervous system. Although they affect millions of patients worldwide and have a profound impact on families and their community, the pathogenic mechanisms underlying these conditions remain unclear. The therapeutic options for patients are scarce and merely palliative. Nevertheless, there is currently great interest among the scientific community in regard to the identification of biomarkers or concrete genetic mutations that might help clinicians anticipate the onset of these diseases in a way to either treat them when still reversible or slow down their development [80]. This is especially relevant given the fact that the protein abnormalities that are characteristic of these diseases are present in patients long before the clinical symptoms become noticeable [81,82].

Most neurodegenerative diseases share in common the accumulation of misfolded proteins; however, they also possess traits associated with progressive neuronal dysfunction and death, such as proteotoxic stress, abnormalities in autophagy, neuroinflammation, apoptosis, ROS production, and mitochondrial dysfunction [83]. Mitochondrial function is of utmost importance for neurons because limited glycolysis causes them to rely exclusively on OXPHOS for energy production. Given that the long neuronal axons require energy transport over long distances, and because synaptic transmission is dependent on calcium signaling, mitochondrial performance needs to be tightly regulated [84]. Taken together, there is compelling evidence to suggest a relevant mitochondrial involvement in the pathogenesis of several neurodegenerative diseases, with their role being particularly significant in Parkinson’s disease (PD). According to several studies, patients of this condition present a selective deficiency of respiratory chain complex I, which is most remarkable in the substantia nigra [85,86]. In Alzheimer’s diseases (AD), it has been demonstrated that patients present impaired oxygen consumption in the brain, further adding to the notion that bioenergetic dysfunction and mitochondrial impairment are common features [87]. In fact, impairment of several mitochondrial enzymes has been detected in AD patients, where respiratory chain complex IV is most affected [88]. Complex IV activity was found to be significantly reduced in the brain tissue of AD patients, which presented a general decrease of electron transport chain function [89]. Apart from AD and PD, mitochondrial perturbations have been reported in many other neurodegenerative diseases, among which amyotrophic lateral sclerosis (ALS) and Huntington’s disease (HD) deserve special mention [90]. Nevertheless, a critical question that remains unanswered is whether mitochondrial dysfunction contributes to the initial stages of neurodegeneration, being primarily responsible for the onset of pathogenesis or just a secondary feature arising from alternative phenomena such as the accumulation of misfolded proteins or cellular stress. Either way, it has been demonstrated that targeting mitochondrial dysfunction is a promising therapeutic strategy for neurodegenerative conditions [91]. 

### 4.1. Parkinson’s Diseases

PD is an age-associated neurodegenerative movement disorder that is mainly caused by the death of dopaminergic neurons in the brain substantia nigra [92]. UPR^mt^ may regulate the occurrence and development of PD. It has been shown that a PINK1 mutation can activate ATFS-1-dependent UPR^mt^ and promote dopaminergic neuron survival in a PD worm model [93]. Ginseng protein protected against neurodegeneration by inducing UPR^mt^ in a PINK1 fly model of PD [94]. Dastidar SG et al. demonstrated that activation of 4E-BP1 correlates with UPR^mt^ induction, which reduces PD associated neurotoxicity in mouse neurons [95]. In fact, mutant LRRK2^G2019S^, the most common PD-causing allele in humans, results in reduced 4E-BP1 function and may contribute to PD pathogenesis by UPR^mt^ underactivation [96]. In addition, altered omi and HtrA2 protein, UPR^mt^-related proteases, causes neuropathy with the same characteristics as PD [97]. 

### 4.2. Alzheimer’s Disease

Alzheimer’s disease (AD) is the most common form of dementia [98]. Clinically, AD is defined by cognitive impairment that is pervasive enough to interfere with a person’s ability to work or complete daily activities. The main histologic characteristics include amyloid plaques that largely contain amyloid beta (Aβ) protein and neurofibrillary tangles (NFTs) that consist of hyperphosphorylated tau protein, both of which cause premature neuronal death. Current AD treatments confer a slight benefit but ultimately do not prevent progression [99].

Mitochondria in AD show numerous changes [100]. First, complex-IV-reduced activity has consistently been observed across numerous tissues. In brain tissue specifically, the decrease in complex IV subunits has been linked with disease progression [101]. Mitochondrial surface area also decreases, cristae structures are altered, and an increased variability in mitochondrial shape has been observed [102]. There are changes in fusion and fission mitochondrial proteins that appear to affect mitochondrial localization in AD neurons [103]. Indeed, neuronal cultures with AD-relevant fission/fusion protein alterations recapitulate AD-like changes in mitochondrial distribution [104].

Compared to cognitively intact controls, the expression of UPR^mt^-related genes is increased by 40–60% in sporadic AD subjects and 70–90% in familial AD, respectively [105]. Sorrentino et al. showed that increasing mitochondrial proteostasis by targeting mitochondrial translation and mitophagy both pharmacologically and genetically increases the fitness and lifespan of worms, cells, and mouse models of AD and reduces amyloid aggregation [106]. In addition, UPR^mt^ is strongly activated and exerts a protective role against Aβ protein toxicity in PITRM1-knockout human cells. In line with this, pharmacological inhibition of UPR^mt^ exacerbates the Aβ proteotoxicity in PITRM1-knockout human cells [107]. Consistent with this study, UPR^mt^ is activated in the mouse brains and human SHSY5Y cells after Aβ treatment, while the inhibition of UPR^mt^ aggravates cytotoxic effects of Aβ [108]. Treatments which increase the NAD+ pool, a sirtuin cofactor, alleviate protein toxicity and improve the memory of AD mouse by activating the UPR^mt^. Taken together, these studies provide the evidence of the protective role of UPR^mt^ on AD. Furthermore, the activation of UPR^mt^ induced by Aβ may depend on the regulation of two signaling pathways: the mevalonic acid pathway and ceramide pathway [109]. Therefore, the activation of UPR^mt^ can relieve the symptoms of AD. 

### 4.3. Huntington’s Disease

Huntington’s disease (HD) is a fatal and inherited neurodegenerative disorder that progresses for 15–20 years after the initial onset [110]. The main cause of HD is the expanded CAG repeats encoding polyglutamine (polyQ) in the N-terminus of the huntingtin (Htt) protein [111]. The genetic cause of HD was identified more than 20 years ago. However, the underlying mechanisms leading to the pathogenesis of HD remain elusive.

Accumulating evidence suggests that mitochondrial dysfunction plays an important role in the pathogenesis of HD [112]. For instance, mutant Htt associates with the outer mitochondrial membrane in different HD models, resulting in mitochondrial permeability transition pore opening, calcium disturbance, reduced ATP production, mitochondrial membrane potential loss, increased ROS production, and premature release of cytochrome c [113]. How mutant Htt can affect selectively medium spiny striatal neurons despite its ubiquitous expression is a key question that still remains unanswered. One of the several theories that have been proposed points again to mitochondria, suggesting that medium spiny neurons, characterized by particularly high-energy demands, are especially susceptible to mutant Huntingtin-induced mitochondrial dysfunction and inhibition of respiration [114]. 

Fu et al. [115] found that mutant Htt suppressed the expression of ABCB10, a component of the UPR^mt^ [116], in various HD models by impairing its mRNA stability. Deletion of ABCB10 induced ROS production and cell death in HD mouse striatal cells. Moreover, ABCB10 was required for CHOP activation under mitochondrial stress. They also showed that chaperone HSP60 and protease Clpp, two downstream genes of CHOP [41], were decreased in HD cells. 

### 4.4. Amyotrophic Lateral Sclerosis

Amyotrophic lateral sclerosis (ALS) is caused by the progressive degeneration of motor neurons in the spinal cord, the brain stem, and the motor cortex. Motor neuron death prompts muscle weakness and paralysis, causing death in 1–5 years from the time of symptoms’ onset. The primary identified ALS-linked gene was superoxide dismutase 1 (SOD1), an antioxidant protein [117]. However, mutations in several other genes have been reported, and various molecular pathways have been associated to neuronal death in ALS [118]. The only two disease-modifying potential therapies currently approved for ALS are riluzole and edaravone, but the exact neuroprotective action of these drugs is unknown, and they present no obvious improvement in ALS patients’ health [119]. The absence of alternative drugs for the treatment of ALS indicates the need for the implementation of novel therapeutic strategies

ALS-associated mitochondrial dysfunction comes in many shapes, including defective OXPHOS, ROS production, impaired calcium buffering capacity, and defective mitochondrial dynamics. In addition, mitochondrial dysfunction appears to be directly or indirectly linked to all mechanisms of toxicity associated with ALS, including excitotoxicity, loss of protein homeostasis, and defective axonal transport [120].

TDP-43 proteinopathy is characterized by the presence of TDP-43 immunoreactive inclusion bodies in the affected tissues. This alteration is present in several neurodegenerative diseases with high predominance in ALS [121,122]. Peng et al. showed that LonP1, one of the key mitochondrial proteases constituting the UPR^mt^ response, protects against TDP-43-induced cytotoxicity and neurodegeneration. They suggest the activation of LonP1 via UPR^mt^ as a treatment for TDP-43 proteinopathy, protecting against or reversing mitochondrial damage as a potential therapeutic approach to these neurodegenerative disorders [123]. Qi et al. demonstrated that the mitochondrial function pathway was disrupted in the brain of SOD1^G93A^ mice, and the replenishment of intracellular NAD+, by providing nicotinamide, could reduce neurotoxic protein aggregates of mitochondria in the brain of SOD1^G93A^ mice [124]. Moreover, they concluded that nicotinamide might modulate mitochondrial proteostasis and improve adult neurogenesis through activation of the UPR^mt^ signaling in the brain of SOD1^G93A^ mice. Curiously, it has been reported that UPR^mt^ is transiently activated in the spinal cord of ALS mice models in the late symptomatic phase. However, there is a significant sex difference in the activation of the UPR^mt^: it was significantly activated in female SOD1^G93A^ mice but not in males [125]. Finally, Gomez et al. hypothesized that SOD1 may have multiple functions: antioxidant, UPR^mt^ regulator, and gene transcription enhancer [126]. Taken together, these functions place SOD1 as a key regulator of the communication between the nucleus and mitochondria.

## 5. Heart Diseases

Maintaining mitochondrial quality in cardiomyocytes is essential, given that they produce 8% of the total ATP consumed by the organism and power the constant contraction and relaxation of the myocardium [127]. In fact, mitochondrial abnormalities are a common feature to all types of cardiomyopathies [128]. Mitochondria with aberrant structures are commonly observed in cardiac cells of all forms of heart disease [129]. Furthermore, cardiomyopathy has been reported due to mitochondrial disease [130], and mitochondrial dysfunction has been linked with hypertension. It is nowadays clear that the importance of functional mitochondria for cardiac health is undeniable. One of the main mechanisms ensuring mitochondrial fitness is proteostasis. It finetunes biogenesis, folding, and degradation of mitochondrial proteins, processes which are commonly altered during cardiac stress. It has been demonstrated that stimulation of UPR^mt^ improves mitochondrial function and reduces cardiac damage in response to stress. In this regard, a recent study proved that UPR^mt^ enhancement with small-molecule agents ameliorates mitochondrial and contractile dysfunction in the murine heart. This is the reason why its authors proposed UPR^mt^ as a potential therapeutic target in heart failure [131]. 

Wang et al. demonstrated the relevance of mitophagy and UPR^mt^ in myocardial injuries and stress [132]. They proposed UPR^mt^ activation with oligomycin, a complex V inhibitor, as a mechanism to reduce sepsis-mediated mitochondrial injury and myocardial dysfunction; however, this cardioprotective effect was imperceptible in mitophagy-inhibited mice. On the other hand, when UPR^mt^ was inhibited, mitophagy-mediated protection of mitochondria and cardiomyocytes was partly blunted. Taken together, their observations suggest that both UPR^mt^ and mitophagy are slightly activated by myocardial stress and that they work together to sustain mitochondrial performance and cardiac function.

## 6. Lifespan

Upregulation of UPR^mt^ has been proposed as the common pathway in lifespan extension induced by mitochondrial defects [133]. Studies in *C. elegans* revealed that the induction of mitochondrial stress at an early age resulted in lifespan extension and UPR^mt^ upregulation which persisted even after the stress inductor was withdrawn [134]. This observation suggests that UPR^mt^ upregulation is directly responsible for lifespan extension in worms. Supporting this idea, Yokoyama et al. demonstrated that transgenic worms expressing the nematode UPR^mt^ protein Hsp70F live over 40% more than wild-type animals [135]. To prove whether UPR^mt^ upregulation also influenced lifespan extension in mammals, a recent study assessed UPR^mt^ activation in a long-lived mouse model, Snell dwarf mice. These mice have a single point mutation in *Pit1* and show more than 40% increase in mean and maximal lifespan [136]. The results of the study show that UPR^mt^ was upregulated in both living mice and cultured cells. Moreover, the authors reported elevated basal and post-stress UPR^mt^ levels in primary fibroblasts derived from Snell mice. However, at the organismal level, UPR^mt^ was upregulated only after doxycycline-induced mitochondrial stress exposure [137]. 

The ability of UPR^mt^ activation to prolong lifespan can be explained by several reasons: its positive impact on mitochondrial function, its ability to reduce mitochondrial metabolic by-products, its modulation of insulin/IGF and mTOR signaling, and its involvement in hormetic cellular maintenance [138]. Following UPR^mt^ activation, mitochondria undergo structural and functional changes. The mitochondrial network becomes fragmented [133], and cellular oxygen consumption is significantly reduced [139,140]. This particular phenotype may be a coordinated response to stress that favors mitophagy and lowers respiration to enable mitochondrial repair. Taken together, these measures would ultimately lead to enhanced homeostasis and cell survival, consequently increasing lifespan.

On top of this, UPR^mt^ is responsible for changes in the production of a series of metabolic by-products with relevant physiological functions. Amidst these, ROS is the most widely studied. It elicits varied cellular stress responses, and by doing so, it influences homeostasis and longevity [141]. Additionally, the UPR^mt^ has an impact on the mevalonate pathway, an important target of statins, and on NAD+, which is the substrate for poly (ADP-ribose) polymerases (PARPs) and sirtuin deacetylases and acts as major longevity regulator [56,142]. The link of UPR^mt^ to longevity might also be explained by its effect on insulin signaling. Reduced insulin/IGF-1 signaling has been directly linked to a series of protective mechanisms such as regulation of endoplasmic reticulum unfolded protein response (UPR^ER^) or activation of heat shock factor 1 (HIF-1), which boosts the expression of small heat shock proteins, leading to lifespan extension in worms [143]. In this line, it has been proven that impairment of the insulin/IGF-1 signaling induces a transient ROS stress signal that potentially activates UPR^mt^ and results in prolonged lifespan in *C. elegans* [144]. Improving the level of cellular NAD^+^ in mice not only increases mitochondrial function but also induces the expression of UPR^mt^-associated genes, preventing skeletal muscle stem cells from aging and enhancing life span [145]. 

Although several different routes have been proposed to explain UPR^mt^ implication in longevity, there is the possibility that its lifespan-prolonging ability is the result of improved cellular housekeeping. It is extensively known that loss of proteostasis and the consequent accumulation of misfolded proteins are a fundamental cause of ageing. UPR^mt^ activation would palliate cellular damage in this context by restoring protein quality control. Not only does it assist correct protein folding via enhancement of chaperones’ expression, but it also promotes refolding of aberrant products and sequestration of protein aggregates in less cytotoxic states [146].

## 7. Therapeutic Concerns

Although UPR^mt^ activation seems to be a promising therapeutic approach for mitochondria- related disorders, it does not come without a risk. Prolonged overactivation of UPR^mt^ can also have severe side effects, as proven by recent studies. Lin et al. demonstrated that, in the context of mtDNA heteroplasmy, activation of the UPR^mt^ and constant ATFS-1 signaling maintain and propagate deleterious mitochondrial genomes in *C. elegans*. This observation led the authors to dispute the role of UPR^mt^ activation in longevity, especially in individuals with a heteroplasmic background, such as inherited mitochondrial mutations or cancer cells. Nevertheless, they point out that it is still to be determined whether UPR^mt^ in mammals acts in a similar way to in *C. elegans* [147].

In this line, Martinez et al. reported a detrimental effect of constitutive UPR^mt^ signaling in *C. elegans* dopaminergic neurons after overexpression or overactivation of ATFS-1. Interestingly, they could not find a clear link between the cell death phenotype observed in dopaminergic neurons expressing ATFS-1 and mitophagy, which in *C. elegans* is primarily regulated by *PINK1*. Knockdown *PINK1* worms did not present signs of neuronal cell death. This led the authors to point out that mitophagy impairment does not seem to be the mechanism responsible for neuronal death elicited by ATFS-1 overactivation. Rotenone treatment, nevertheless, had a neuroprotective attenuation effect dependent on *PINK1*. Overall, these results indicate that ATFS-1 overexpression might promote the accumulation of defective mitochondria in dopaminergic neurons, leading to abnormal cellular homeostasis and cell death. The phenotype can be reversed by increasing aberrant mitochondria clearance via mitophagy [148].

Regarding aging, most studies agree that UPR^mt^ contributes to health and longevity [133], including neurons where UPR^mt^ can activate protective cell non-autonomous signals and epigenetically rewire animal models to live longer [149]. However, in some circumstances, UPR^mt^ seems to promote aging, as pointed out by Angeli S et al. [150]. They discovered in *C. elegans* that certain mitochondrial insults during development lead to the lasting activation of the UPR^mt^ and are associated with longevity; however, the same stimuli during adulthood induce premature aging. In this case, suppression of the UPR^mt^ via genetic or pharmacological interventions is protective. These results, again, evidence the complexity of UPR^mt^ pathways. 

Constitutive UPR^mt^ activation can also be extremely detrimental for cancer patients. It is widely known that cancer cells make use of numerous stress response pathways to counteract and survive endogenous, exogenous, and environmental stresses. Among them, UPR^mt^ may help cancer cells clear excessive cellular damage that could eventually lead to apoptosis. Similarly to cardiomyocytes, cancer cells use this stress response to ensure their survival and proliferation [151]. In fact, HSP60, a key chaperone of UPR^mt^ activation process across different species, is overexpressed in several cancer types such as acute myeloid leukemia, pancreatic ductal adenocarcinoma, ovarian carcinoma, breast ductal invasive carcinoma, prostate adenocarcinoma, and others [48,152]. By folding and refolding oncoproteins and denatured/misfolded proteins within mitochondria, UPR^mt^ facilitates cancer growth and increases the apoptotic threshold of cancer cells [153]. To better understand the implication of UPR^mt^ and HSP60 in cancer, Tsai et al. proved that HSP60 overexpression significantly increases cellular migration and invasion in vitro, as well as increased tumor volume and metastasis in vivo [154]. Moreover, it has been reported that HSP60 plays a crucial role in the metastasis of pancreatic cancer cells [155] and increases motility in breast cancer cells [156]. Thus, in cancer cells, the UPR^mt^ is exploited for mitochondrial repair and tumor growth, invasion, and metastasis promotion. In fact, disrupting proteostasis in cancer cells by targeting UPR^mt^ constitutes a novel anticancer therapeutic strategy [157]. In addition, cancer stem cells (CSCs) represent a highly tumorigenic group of cells in primary tumors [158]. Mitochondrial changes in CSCs, including morphological changes, abnormal activation of signaling pathways, mitochondrial dysfunctions, production of ROS, enhanced mitophagy, and UPR^mt^ modulation, are key regulators of CSC proliferation and apoptosis and are also among the reasons for the failure of antitumor treatments [159]. Therefore, targeting UPRmt in CSCs can be essential for the effective treatment of cancer [160]. 

## 8. Conclusions

The potential of UPR^mt^ activation therapies is undeniable. The UPR^mt^ is a fundamental stress response that improves cellular fitness and proteostasis, both of which are leading causes of very diverse conditions when impaired. From mitochondrial diseases to neurodegeneration or cardiovascular disorders, mitochondrial dysfunction has been identified as a pivotal factor for disease development (Table 1). Studies have demonstrated the therapeutic efficacy of UPR^mt^, yet no specifical activators of UPR^mt^ have been applied for patient treatment so far. The side effects and deleterious repercussions of the sustained activation of UPR^mt^ in patients are important concerns for clinicians. Therefore, efforts have been made to identify safe, well-studied drugs that succeed in activating this mitochondrial response. Nevertheless, further studies tackling the safety profile of UPR^mt^ activators and their possible long-term side effects need to be conducted to ascertain that patients will not undergo risks when adopting these novel therapeutic approaches. 

## Figures and Tables

**Figure 1 biomedicines-10-01611-f001:**
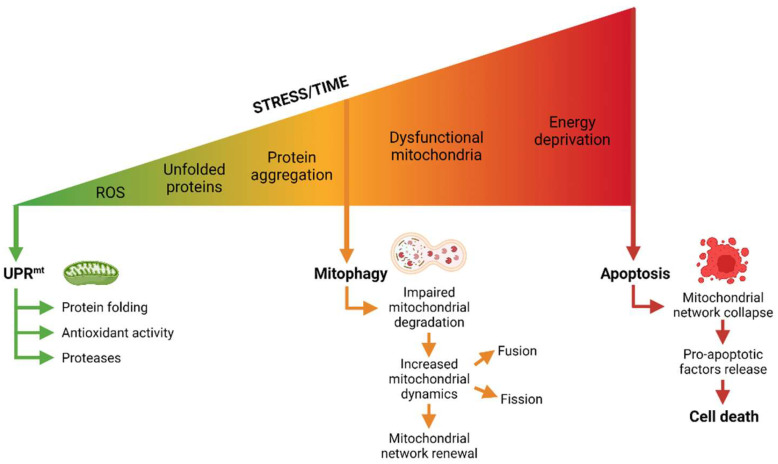
Mitochondrial stress response. The most common mitochondrial stressors are ROS, unfolded proteins, and protein aggregates. To deal with such insults and their consequences, mitochondria possess several mechanisms: First, the UPR^mt^ will promote the expression of antioxidant proteins, chaperones, or proteases. If damage persists and/or increases, the cell will try to remove highly damaged mitochondria with mitophagic processes. Mitophagy degrade dysfunctional mitochondria by a combination of mitochondrial fission and autophagy; then mitochondrial biogenesis and fusion are enhanced to restore energetic balance. All of these quality-control mechanisms are non-exclusives, and cells can regulate them according to the situation. Finally, if the cell is unable to overcome the energetic crisis, it will undergo apoptosis through mitochondrial release of pro-apoptotic factors. Figure was created with BioRender (BioRender.com).

**Figure 2 biomedicines-10-01611-f002:**
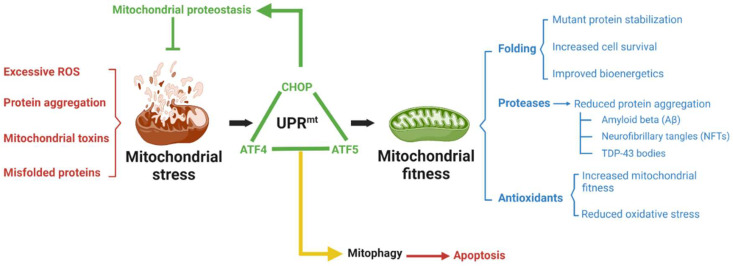
The UPR^mt^ is a conserved adaptive mechanism for improving cell survival under mitochondrial stress. The UPR^mt^ may act as a first line of defense against mitochondrial stress and involves communication between the stressed mitochondria and the nucleus. Activation of the UPR^mt^ aims to restore protein homeostasis and function within the mitochondria, therefore preserving cellular functions. UPR^mt^ is mainly regulated for a triad of proteins—ATF4, ATF5, and CHOP; however, although their interactions are not well stablished yet. Different reports have described the protective effects of UPR^mt^ activation in various disease conditions, such as mitochondrial diseases, neurodegeneration, or heart diseases. However, after trespassing a certain stress threshold and to prevent the toxic effects of dysfunctional cell processes, affected mitochondria are eliminated via mitophagy, which ultimately will lead to apoptosis if the damage is irreversible. Figure was created with BioRender (BioRender.com).

**Table 1 biomedicines-10-01611-t001:** Summary of UPR^mt^-related conditions and studies.

Condition	Related Studies
Mitochondrial diseases	[60,64,73]
Parkinson’s disease	[93,94,95,96,148]
Alzheimer’s disease	[105,106,107,108,109]
Huntington’s disease	[115]
Amyotrophic lateral sclerosis	[121,122,123,124]
Heart diseases	[132]
Aging	[139,140,141,142]
Cancer	[157,160]

## Data Availability

Not applicable.

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
