# Peer review of "Activation of the Mitochondrial Unfolded Protein Response: A New Therapeutic Target?"

_biomedicines, 2022, doi:10.3390/biomedicines10071611_

Round 1
Reviewer 1 Report
the work does not present any image, for example one of the mitochondrion or proteins. You can try to explain it better to the author through a graphical abstract. The pages seem interesting to the reader, however I would add some guiding notes to the eye. Having made these corrections, I believe the paper can be accepted.Author Response
Reviewer1
The work does not present any image, for example one of the mitochondrion or proteins. You can try to explain it better to the author through a graphical abstract. The pages seem interesting to the reader, however I would add some guiding notes to the eye. Having made these corrections, I believe the paper can be accepted.
Tables and figures have been added to the main text following your suggestion.
Reviewer 2 Report
Type of manuscript: Review
Title: Activation of the Mitochondrial Unfolded Protein Response: A new therapeutic target
In this manuscript, the authors summarized the association of mitochondrial unfolded protein response with various diseases and proposed that targeting mitochondrial unfolded protein response might be a promising therapeutic strategy. This is an interesting topic. However, the present form lacks a mechanistic view of the underlying mechanism. Suggestions to improve the quality of the manuscript is listed below.
Specific Comments:
1. The authors have listed previous observations regarding the association of mitochondrial unfolded protein response with various diseases. However, the present form did not distinguish whether mitochondrial unfolded protein response is the cause or consequence of these diseases. Say, pathological conditions could also be considered as stressed conditions. Is mitochondrial unfolded protein response a one of the consequences or a companied event of the progressed disease, or contribute to the disease progression?
2. The authors should take caution of results of regulating one specific gene. Up- or down-regulation of one housekeeping gene may produce numerous changes of the cells including the mentioned mitochondrial unfolded protein response. However, the contribution of mitochondrial unfolded protein response pathway should be carefully evaluated.
3. The manuscript lacks schematic figures to show either the molecular basis of mitochondrial unfolded protein response in various diseases or potential mechanisms of targeting mitochondrial unfolded protein response.
4. Lines 224 to 262 can be shortened since some have been addressed in the following subsections. Section 7 is written by listing several studies. I suggest the authors to strengthen this section by re-organizing related studies in a mechanistic way. Furthermore, from the title of this manuscript, this section should be the key part of this manuscript.
Author Response
Reviewer 2
Type of manuscript: Review
Title: Activation of the Mitochondrial Unfolded Protein Response: A new therapeutic target
In this manuscript, the authors summarized the association of mitochondrial unfolded protein response with various diseases and proposed that targeting mitochondrial unfolded protein response might be a promising therapeutic strategy. This is an interesting topic. However, the present form lacks a mechanistic view of the underlying mechanism. Suggestions to improve the quality of the manuscript is listed below.
Specific Comments:
- The authors have listed previous observations regarding the association of mitochondrial unfolded protein response with various diseases. However, the present form did not distinguish whether mitochondrial unfolded protein response is the cause or consequence of these diseases. Say, pathological conditions could also be considered as stressed conditions. Is mitochondrial unfolded protein response a one of the consequences or a companied event of the progressed disease, or contribute to the disease progression?
UPRmt activation is a consequence derived from mild mitochondrial stress and therefore present in many mitochondrial-related diseases. In this review, we focus on the role of UPRmt in those diseases. Furthermore, we present several cases where impaired UPRmt could lead or aggravate disease pathophysiology. On the other hand, we highlight several studies that suggest that UPRmt activation could be a potential treatment.
- The authors should take caution of results of regulating one specific gene. Up- or down-regulation of one housekeeping gene may produce numerous changes of the cells including the mentioned mitochondrial unfolded protein response. However, the contribution of mitochondrial unfolded protein response pathway should be carefully evaluated.
Thank you for the comment. We completely agree. The following sentence has been included in the text:
However, the contribution of particular genes in the UPRmt response pathway should be carefully evaluated, taking into account that many of them are constitutive genes that are required for the maintenance of basic cellular function, and their up- or down-regulation may produce numerous changes in the cells including the mentioned UPRmt.
- The manuscript lacks schematic figures to show either the molecular basis of mitochondrial unfolded protein response in various diseases or potential mechanisms of targeting mitochondrial unfolded protein response.
A figure has been included in the text following your suggestion.
- Lines 224 to 262 can be shortened since some have been addressed in the following subsections. Section 7 is written by listing several studies. I suggest the authors to strengthen this section by re-organizing related studies in a mechanistic way. Furthermore, from the title of this manuscript, this section should be the key part of this manuscript.
The text has been modified according to your advice.
Reviewer 3 Report
The manuscript (ID: biomedicines-1757596) has been submitted by the authors Juan M. Suárez-Rivero, Carmen Julia Pastor-Maldonado, Suleva Povea-Cabello, Mónica Álvarez-Córdoba, Irene Villalón-García, Marta Talaverón-Rey, Alejandra Suárez Carrillo, Manuel Munuera-Cabeza, Diana Reche-López, Paula Cilleros-Holgado, Rocío Piñero-Pérez and José A. Sanchez-Alcazar as a review to the section Neurobiology and Neurologic Disease of the scientific journal Biomedicines. The manuscript is entitled „Activation of the Mitochondrial Unfolded Protein Response: A new therapeutic target”. Although, the authors discuss unfolded proteins there are no structural data discussed. Furthermore, I am missing instructive figures and tables. The draft is considered as a review article. Unfortunately, actual publications are not cited:
Wang et al. Cell & Bioscience (2022) Insight into the mitochondrial unfolded protein response and cancer: opportunities and challenges 12:18
https://doi.org/10.1186/s13578-022-00747-0
Gu LF, Chen JQ, Lin QY, Yang YZ. Roles of mitochondrial unfolded protein response in mammalian stem cells. World J Stem Cells 2021; 13(7): 737-752 https://dx.doi.org/10.4252/wjsc.v13.i7.737
After a major revision the draft could become a valuable contribution for the corresponding section of the target journal.
Author Response
Reviewer 3
The manuscript (ID: biomedicines-1757596) has been submitted by the authors Juan M. Suárez-Rivero, Carmen Julia Pastor-Maldonado, Suleva Povea-Cabello, Mónica Álvarez-Córdoba, Irene Villalón-García, Marta Talaverón-Rey, Alejandra Suárez Carrillo, Manuel Munuera-Cabeza, Diana Reche-López, Paula Cilleros-Holgado, Rocío Piñero-Pérez and José A. Sanchez-Alcazar as a review to the section Neurobiology and Neurologic Disease of the scientific journal Biomedicines. The manuscript is entitled „Activation of the Mitochondrial Unfolded Protein Response: A new therapeutic target”. Although, the authors discuss unfolded proteins there are no structural data discussed. Furthermore, I am missing instructive figures and tables.
The main objective of the review is to provide an update the UPRmt response from the therapeutic point of view. For a detailed description of molecular mechanisms and structural data several recent reviews are recommended.
Tables and figures have been added to the main text following your suggestion.
The draft is considered as a review article. Unfortunately, actual publications are not cited:
The recent publications have been included in the text following your suggestion.
Wang et al. Cell & Bioscience (2022) Insight into the mitochondrial unfolded protein response and cancer: opportunities and challenges 12:18
https://doi.org/10.1186/s13578-022-00747-0
Gu LF, Chen JQ, Lin QY, Yang YZ. Roles of mitochondrial unfolded protein response in mammalian stem cells. World J Stem Cells 2021; 13(7): 737-752 https://dx.doi.org/10.4252/wjsc.v13.i7.737
After a major revision the draft could become a valuable contribution for the corresponding section of the target journal.
Round 2
Reviewer 1 Report
The work si well written.In my opinioni it can ben accepted.
Author Response
Thank you for your comments
Reviewer 2 Report
The authors have addressed most of my concerns. However, the quality of the manuscript could be improved if considering the following issues (some have been raised in my previous review).
1, Only one figure and one table are added. More figures are expected to improve the clarity and readability of the review.
2, I suggest that a question mark could be included in the title of the manuscript since there are clinical concerns regarding the topic. Say, "Activation of the Mitochondrial Unfolded Protein Response: A new therapeutic target?"
Author Response
1, Only one figure and one table are added. More figures are expected to improve the clarity and readability of the review.
An additional figure has been added accordingly
2, I suggest that a question mark could be included in the title of the manuscript since there are clinical concerns regarding the topic. Say, "Activation of the Mitochondrial Unfolded Protein Response: A new therapeutic target?"
We agree with reviewer‘s suggestion
Reviewer 3 Report
The authors have followed all suggestions and comments. The draft can now be published.
Author Response
Thank you for your comments